# Climate Factors Influence Above- and Belowground Biomass Allocations in Alpine Meadows and Desert Steppes through Alterations in Soil Nutrient Availability

**DOI:** 10.3390/plants13050727

**Published:** 2024-03-04

**Authors:** Jiangfeng Wang, Xing Zhang, Ru Wang, Mengyao Yu, Xiaohong Chen, Chenghao Zhu, Jinlong Shang, Jie Gao

**Affiliations:** 1College of Life Sciences, Xinjiang Normal University, Urumqi 830054, China; wjf2088683747@163.com (J.W.); zxyybh@163.com (X.Z.); wangruavon@163.com (R.W.); yao02292023@163.com (M.Y.); cheng1352023@163.com (X.C.); 2East China Survey and Planning Institute, National Forestry and Grassland Administration, Hangzhou 430010, China; zchmee@126.com; 3Key Laboratory of Earth Surface Processes of Ministry of Education, College of Urban and Environmental Sciences, Peking University, Beijing 100871, China

**Keywords:** above–belowground biomass, alpine meadows, desert steppes, climate factors, soil nutrients

## Abstract

Biomass is a direct reflection of community productivity, and the allocation of aboveground and belowground biomass is a survival strategy formed by the long-term adaptation of plants to environmental changes. However, under global changes, the patterns of aboveground–belowground biomass allocations and their controlling factors in different types of grasslands are still unclear. Based on the biomass data of 182 grasslands, including 17 alpine meadows (AMs) and 21 desert steppes (DSs), this study investigates the spatial distribution of the belowground biomass allocation proportion (BGBP) in different types of grasslands and their main controlling factors. The research results show that the BGBP of AMs is significantly higher than that of DSs (*p* < 0.05). The BGBP of AMs significantly decreases with increasing mean annual temperature (MAT) and mean annual precipitation (MAP) (*p* < 0.05), while it significantly increases with increasing soil nitrogen content (N), soil phosphorus content (P), and soil pH (*p* < 0.05). The BGBP of DSs significantly decreases with increasing MAP (*p* < 0.05), while it significantly increases with increasing soil phosphorus content (P) and soil pH (*p* < 0.05). The random forest model indicates that soil pH is the most important factor affecting the BGBP of both AMs and DSs. Climate-related factors were identified as key drivers shaping the spatial distribution patterns of BGBP by exerting an influence on soil nutrient availability. Climate and soil factors exert influences not only on grassland biomass allocation directly, but also indirectly by impacting the availability of soil nutrients.

## 1. Introduction

Grasslands play a crucial role as carbon sinks within terrestrial ecosystems, making a significant contribution to the global carbon cycle [1]. The distribution of plant biomass between aboveground and belowground components reflects the adaptive survival strategies of plants in diverse habitats, developed over extended periods in response to environmental changes and stresses [2]. The distribution of aboveground and belowground biomass in grassland ecosystems is widely discussed, and in general, belowground biomass is higher than aboveground biomass in grassland ecosystems [3,4]. While extensive research has been conducted on aboveground plant biomass [5,6], the allocation of belowground biomass and its controlling factors remain relatively underexplored due to the challenges associated with obtaining belowground biomass data [7]. Investigating the allocation strategies of grassland plants for above- and belowground biomass, along with their governing factors, can advance our understanding of carbon allocation and storage dynamics within grassland ecosystems [8].

Previous research has established that the allocation of plant biomass between aboveground and belowground components is primarily influenced by climatic and soil nutrient factors [9]. According to the optimal partitioning hypothesis, plants in resource-limited environments adjust their biomass allocation between above- and belowground components to adapt to environmental stresses [10]. The effects of climate change and soil nutrient availability can impact plant investment in belowground components within grassland ecosystems [11,12,13].

Temperature and precipitation are pivotal climatic factors [14]. Existing studies demonstrate that the interaction between temperature and moisture significantly impacts plant belowground biomass allocation, particularly in arid regions [15]. In arid and semi-arid grasslands, elevated temperatures influence enzyme reaction rates and reduce the photosynthetic capacity of aboveground plant components [16]. Concurrently, higher temperatures accelerate the evaporation of surface water from the soil, diminishing soil moisture availability and, consequently, the plant’s ability to access soil nutrients [17]. Consequently, in arid regions, plants allocate more resources to root biomass to secure water and adapt to drought stress [18].

Soil nutrients, including soil nitrogen content, soil phosphorus content, and soil pH, exert a significant influence on the BGBP [19,20]. Research suggests that soil nutrients are the primary limiting or influential factors in shaping BGBP in alpine grasslands [21]. Elevated levels of soil nitrogen and phosphorus stimulate the allocation of a higher proportion of plant biomass to roots, enabling increased energy investment in root growth [22]. Soil pH serves as an indicator of soil fertility and is closely associated with soil microbial activity [23,24]. Within acidic soil, adjusting the soil pH within a certain range can enhance soil microbial activity, facilitating soil nutrient transformations and supply, consequently leading to an increase in BGBP [25].

In addition to environmental factors, grassland productivity also significantly impacts the BGBP [26]. There is a strong positive correlation between aboveground biomass and net primary productivity (NPP), with biomass serving as the primary driver of NPP [27]. Grassland productivity reflects the amount of organic carbon fixed through photosynthesis in aboveground plant parts [28]. Therefore, as the NPP gradually increases in a specific grassland area, it signifies that the vegetation in that area acquires more biomass by increasing the aboveground growth, resulting in a decrease in the BGBP for the vegetation in that area [29].

Temperature, precipitation, and soil nutrients have the capacity to alter plant biomass allocation strategies [9]. However, there is a scarcity of studies that specifically examine the differences in belowground biomass allocation between desert steppes (DSs) and alpine meadows (AMs) [30]. Owing to variations in geographical location, hydrothermal conditions, and soil nutrient profiles, the distribution of belowground biomass significantly differs between an AM and DS [11,12,13]. High-elevation grasslands are predominantly found in mountainous and plateau regions, characterized by dry and cold climates, and prolonged snow cover periods. Soil nutrient availability is relatively limited in these high-elevation areas [31], making soil nutrient availability a critical environmental factor influencing plant belowground biomass allocation in these grasslands. Conversely, desert grasslands are primarily situated in arid and semi-arid regions, characterized by low precipitation and high rates of water evaporation [5]. Therefore, climatic factors may be the primary environmental determinants influencing plant belowground biomass allocation in desert grasslands. In essence, the distribution of belowground biomass in grasslands is influenced by climate factors and soil nutrients [9]. Climate factors impact soil nutrients through processes like weathering, leaching, and biological interactions, subsequently influencing belowground biomass allocations in both AMs and DSs through their effects on the soil environment [32,33].

It remains unclear whether significant differences exist in the spatial distribution of the BGBP among various grassland types, and whether the dominant environmental factors and specific ecological processes responsible for the spatial distribution of BGBP are consistent. To address this question, we propose the following hypotheses based on biomass data from 182 plots spanning 17 AMs and 21 DSs in China: (1) substantial variations are observed in the spatial distributions of plant BGBP values across distinct grassland types; (2) climate factors are the primary environmental determinants influencing the BGBP in DSs, while soil nutrient factors play a pivotal role in shaping the BGBP in AMs; and (3) climate factors influence the spatial distribution patterns of BGBP in different grassland types by modulating the availability of soil nutrients.

## 2. Results

Distinct spatial disparities are evident in the BGBPs between AMs and DSs, with AMs exhibiting a significantly higher BGBP. As the mean annual temperature (MAT) increases, the BGBP of the AM significantly declines (*p* < 0.001) (Figure 1A). Similarly, with rising the mean annual precipitation (MAP), both the AM and DS show a significant decrease in the BGBP (*p* = 0.006, *p* < 0.001) (Figure 1B). Notably, the MAP is more effective in explaining the spatial variation in the BGBP in the DS compared to the AM, as indicated by higher R^2^ values (R^2^ = 0.13, *p* < 0.001; R^2^ = 0.08, *p* = 0.006) (Figure 1B).

The BGBPs of both the AM and DS exhibit significant increases with rising soil P and pH (*p* < 0.05) (Figure 2B,C). Soil N displays a significant positive correlation with the BGBP of the AM (R^2^ = 0.34, *p* < 0.001) (Figure 2A). Among these soil factors, soil pH has the most pronounced impact on the AM’s BGBP and emerges as the primary driver of its variation (R^2^ = 0.78, *p* < 0.001) (Figure 2C). In contrast, the net primary productivity (NPP) of the AM is significantly negatively correlated with the BGBP (R^2^ = 0.22, *p* < 0.001), while the NPP in the DS does not exhibit a significant correlation with the BGBP (*p* > 0.05) (Figure 3).

Climatic factors, soil nutrient factors, and NPP values are crucial factors that affect the spatial distribution of BGBP values for both AMs and DSs, and they have significant correlations with each other (Figure 4). Climatic factors have a negative effect on the BGBP of AMs but a positive effect on the BGBP of DSs, while soil nutrient factors have positive effects on the BGBP values of both AMs and DSs. Additionally, NPP has a negative effect on BGBP in AMs but a positive effect on BGBP in DSs (Figure 5A,B).

Climatic factors can have a direct and significant impact on BGBP (*p* < 0.05) (Figure 6). However, they can also significantly affect BGBP by influencing the availability of soil nutrients (*p* < 0.05) (Figure 6). Overall, the indirect impact of climatic factors on BGBP is stronger than the direct impact.

## 3. Discussion

Distinct geographical disparities are evident in the BGBP between the AM and DS, with the AM exhibiting a significantly higher BGBP. This observation aligns with our previous hypothesis and is consistent with the findings from other studies [34]. Different vegetation types exhibit varying responses to climate change [35]. DSs are typically found in arid plain regions, where rainfall serves as the primary limiting factor for plant growth [36]. To adapt to the impact of drought stress on plant growth and development, plants allocate a greater proportion of organic matter to their underground parts [37]. In contrast, the elevated BGBP in an AM compared to a DS is attributed to its distribution in cold, high-altitude regions with not only cold temperatures, but also reduced rainfall. In such environments, grass plants tend to increase their investment in root development as a response to the dual environmental stresses [38].

Numerous studies have consistently shown a strong correlation between the BGBP of plants and climate factors, particularly temperature and precipitation [39,40]. In line with the Optimal Allocation Hypothesis [41,42], a temperature increase within a certain range effectively boosts enzyme activity [43]. To capture more light energy and enhance their photosynthetic capacity, plants allocate a greater proportion of organic matter to aboveground structures, consequently reducing the allocation to belowground active processes [44,45]. This results in a significant decrease in BGBP for AMs with rising temperatures. Notably, DSs show reduced sensitivity to temperature changes, indicating that temperature is not the primary limiting factor influencing DS biomass allocation.

Both AMs and DSs exhibit a significant decline in BGBP values with increased rainfall, primarily due to the correlation between root size and plant water and nutrient absorption capacity. In water-scarce grasslands, plants respond by increasing both horizontal and vertical root growth to access more water resources [46]. Consequently, they gradually reduce their investment in belowground root construction, leading to a decrease in BGBP [47]. DSs display higher sensitivity to rainfall changes compared to AMs. Plants in arid regions have adapted to limited water resources over time, relying heavily on water efficiency for survival and growth [12]. However, their biomass allocation is more responsive to fluctuations in rainfall patterns, including increases or decreases in precipitation [48]. In contrast, AM plants thrive in relatively humid environments and have a lower absolute water dependency [13].

Soil plays a pivotal role in supplying essential nutrients for plant growth and development, thus exerting a significant impact on plant biomass allocation strategies [12]. The BGBP of AMs exhibited a notable increase with rising soil nitrogen and phosphorus contents. In alpine regions, the lower temperatures limit soil microbial activity, resulting in slower soil organic matter decomposition and organic matter accumulation. Consequently, the higher soil organic matter content in alpine regions reflects, to some extent, a reduced resource utilization capacity [49]. With increased soil nitrogen and phosphorus contents, soil nutrient availability decreases significantly, prompting plants to allocate more resources to root systems for enhanced soil resource acquisition [50]. This increased root biomass allows plants to explore a larger soil volume for nutrient uptake, leading to higher BGBP levels [51].

In general, acidic soil tends to exhibit greater soil nutrient availability compared to alkaline soil. Therefore, as soil pH increases (shifting from acidic to alkaline), the stress related to soil resource availability intensifies, resulting in higher BGBP levels [52]. In contrast, DSs display lower sensitivity to soil nutrient changes. Desert plants tend to adopt conservative growth strategies that they maintain even when nutrient availability increases, as water often serves as the more critical limiting factor. This strategy reduces their dependence on and sensitivity to nutrient changes [47]. In contrast, plants in AMs may rely more on soil nutrients due to the need for rapid growth and reproduction within a short growing season, rendering them more sensitive to shifts in the soil nutrient status [21].

In our research findings, net primary productivity (NPP) plays a crucial role in structural equation models (SEMs), revealing a negative correlation between BGBP and grassland NPP (Figure 6). As the NPP increases, plant photosynthesis generally intensifies, necessitating a larger leaf area to capture sunlight [29,53]. Consequently, plants allocate a greater portion of their biomass to aboveground structures to support increased leaf areas and faster growth rates [54]. Moreover, high-productivity environments often signify lower environmental stress, encouraging plants to allocate more resources to belowground biomass as a strategy to withstand stress [26]. In less stressful conditions, plants may not heavily rely on these defense strategies. As productivity rises, grassland species composition may undergo changes [55]. Certain plant species favoring fertile soils may become more prevalent. These plants typically invest more resources in aboveground growth than in belowground biomass [56]. Consequently, the BGBP of grasslands exhibits a significant decline with the increasing NPP.

The interplay between climate and soil factors can significantly impact plant biomass allocation strategies [39,57,58]. Our structural equation model (SEM) results demonstrate that climate factors and soil nutrient factors exert both direct and indirect effects on grassland biomass allocation. Climate factors, such as temperature and precipitation, play a direct role in determining water and temperature stress levels, which subsequently influence the allocation of plant resources, favoring root system investment under growth and survival pressures [39]. Additionally, climate change impacts the physical and chemical properties of the soil, including nutrient availability and microbial activity, integral components of nutrient cycling. These changes further affect plant biomass allocation strategies, with resource-rich environments promoting aboveground growth for reproduction and competitiveness, while resource-limited environments encourage increased subsurface biomass for improved access to water and nutrients [51].

Consequently, climate change not only directly influences plant physiological responses, but also indirectly shapes grassland BGBP through soil nutrient dynamics regulation [8,9]. As global climate conditions become hotter and water scarcity increases, plants allocate more biomass to the belowground portion. Our study not only elucidates the regulatory impacts of climate change and soil nutrients on grassland BGBP, but also unveils the response mechanisms of grassland plant biomass allocation to global warming.

## 4. Conclusions

Utilizing the biomass data from 182 plots across 17 alpine meadows (AMs) and 21 desert steppes (DSs) in China, this study examined the influence of climate factors, soil nutrients, and net primary productivity (NPP) on the BGBP across various grassland types. The findings reveal significant spatial distribution differences in BGBP values among the grassland types. Climate-related factors, by modulating soil nutrient availability, emerged as the primary determinants of the BGBP spatial distribution patterns. Consequently, emphasizing the impact of climate change on the allocation of biomass above- and belowground in grasslands is essential for forecasting terrestrial ecosystems’ reactions to global climate change. Additionally, investigating the allocation strategies and influential factors on grassland plants’ above- and belowground biomass contributes to a deeper understanding of carbon allocation and storage dynamics within grassland ecosystems. This knowledge is crucial for accurately predicting carbon feedback on a regional scale in the future.

## 5. Material and Methods

### 5.1. Sample Plots and BGBP Data

Most grasslands in China are situated in the arid and semi-arid regions of Northern China and Tibetan Plateau [59]. In this study, we investigated the spatial distribution patterns and influencing factors of BGBP in different grassland types in China (Figure 7A).

In order to reveal the relationship between climatic factors and soil factors on the belowground biomass of grasslands, the grassland communities far away from human disturbance were selected. Consider comprehensively the distribution characteristics of grassland types and grassland vegetation at each research point. In order to avoid the non-independence of sample plot data, at least four 10 m × 10 m sample plots with typical regional vegetation were randomly selected at each point, and the precise geographical information of each sample plot, such as longitude and latitude, was recorded. Set up 3 sample quadrats with a size of 1 m × 1 m. The root system was collected by the root excavation method in each sample plot, and the digging depth was 50 cm. When digging roots, care should be taken not to damage the roots, and the entire root system should be dug out as much as possible, including taproots, lateral roots, and fine roots. The leaves and stems of the aboveground part of the plant were harvested by the harvest method. All plant samples above- and belowground were cured at 120 °C for two hours and then dried at 70 °C until they reached a constant weight; the samples were then weighed using a 0.0001 g balance.
Aboveground biomass (AGB) = stem dry matter weight + leaf dry matter weight.
Belowground biomass (BGB) = Root dry weight
Belowground biomass proportion (BGBP) = BGB/(AGB + BGB) × 100%

### 5.2. Environmental Data

The mean annual temperature (MAT) and mean annual precipitation (MAP) for each site were extracted from the WorldClim (version 2.0) database (https://worldclim.org, last accessed on 10 October 2022) at a spatial resolution of 1 km.

Soil pH, N, and P within the uppermost 30 cm of soil were obtained from https://www.csdn.store (accessed on 10 October 2022) and https://www.osgeo.cn/data/wc137 (accessed on 10 October 2022). In the abovementioned data acquisition website, soil N concentrations of all samples were measured by a C–N analyzer (PE-2400 II; Perkin-Elmer, Boston, MA, USA), while soil P concentrations were measured using the molybdate–ascorbic acid method after H_2_SO_4_–H_2_O_2_ digestion [60]. The soil pH was determined in a 1:2.5 soil/water solution using a pH meter [61].

### 5.3. NPP Data

The grassland NPP data were obtained from NASA with a resolution of 250 × 250 m (https://search.earthdata.nasa.gov/search, last accessed 10 October 2022). The Carnegie Ames Stanford Approach (CASA) model was used to estimate the NPP, following the method described by Du et al. (2022) [62]:(1)NPP(x,t)=APAR(x,t)×ε(x,t)
where APAR(x,t) represents the photosynthetically active radiation (PAR, in units of MJ/m^2^) absorbed at pixel x in month t, and ε(x, t) represents the actual light-energy utilization at pixel x in month t (g C/MJ).

### 5.4. Statistical Analyses

Significance tests were conducted at a significance level of 0.05 to evaluate the differences in BGBP values between the AM and DS. The R package “agricolae” (version 4.1.0, R Core Team, 2020) was used for these significance tests [57]. In order to study the impact of biotic and abiotic factors on the spatial variation in BGBP and consider the impact of random effects on biomass allocation, we constructed linear regression models and nonlinear regression models. Finally, we selected the linear mixed-effects model with the lowest Akaike information criterion (AIC) value. The goodness of fit of the model was evaluated using R^2^. A linear mixed-effects analysis was performed using the R package “lme4”. To analyze the effects of climate factors, soil factors, and NPP on BGBP, the natural logarithm of the response ratio (LnRR) was used [63]. Environmental factors were divided into experimental and control groups, with the experimental group consisting of samples above the mean and the control group consisting of samples below the mean. The response ratio (RR) was the ratio of BGBP between the experimental group (Xt) and the control group (Xc). A logarithmic transformation was used to facilitate the statistical analysis.
(2)$$\mathrm{LnRR}=\frac{\bar{X_{t}}}{\bar{X_{c}}}=\mathrm{Ln}\bar{X_{t}}-\mathrm{Ln}\bar{X_{c}}$$

We utilized a random-effects model in our analysis to compute the effect sizes (LnRRs) for both high and low levels of environmental factors, soil factors, NPP, and BGBP. All statistical analyses were carried out in R, and the meta-analysis was performed using the metafor package [64]. The results were then presented in a forest plot. Furthermore, we conducted a correlation analysis between different environmental factors, soil factors, NPP, and BGBP using the Mantel test method in the “vegan” package of R language [65]. We also plotted the correlation heatmap between each factor and BGBP.

To determine whether climate and soil nutrient factors had a direct or indirect impact on BGBP through community characteristics, we created two structural equation models (SEMs) [66]. These SEMs were built using the R package “piecewiseSEM”. The models assumed that: (1) climate factors affected soil nutrient factors, and both climate factors and soil nutrient factors together affected BGBP, and (2) climate factors had a direct effect on BGBP.

## Figures and Tables

**Figure 1 plants-13-00727-f001:**
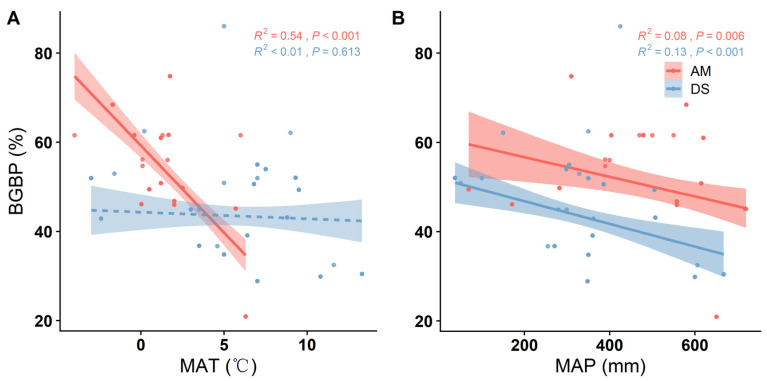
The linear relationship between climate factors and belowground biomass production (BGBP) of alpine meadows and desert steppes. The R^2^ value represents the goodness of fit of the model, and the *p*-values indicate the significance of the results. The climate factors considered are: (**A**) mean annual temperature (MAT) and (**B**) mean annual precipitation (MAP).

**Figure 2 plants-13-00727-f002:**
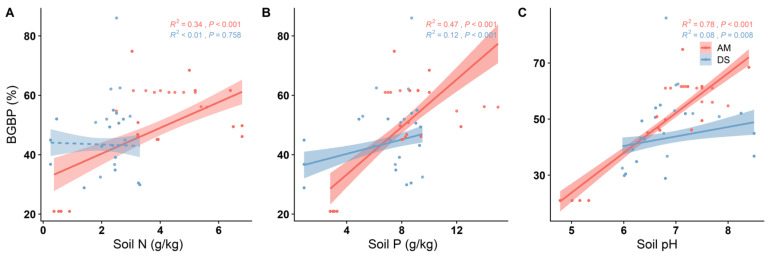
Linear relationship between soil factors and BGBP (belowground biomass production) of alpine meadows and desert steppes. R^2^ represents the goodness of fit of the model, and *p*-values indicate significance. Soil factors include: (**A**) soil total nitrogen content (Soil N); (**B**) soil available phosphorus content (Soil P); (**C**) soil pH.

**Figure 3 plants-13-00727-f003:**
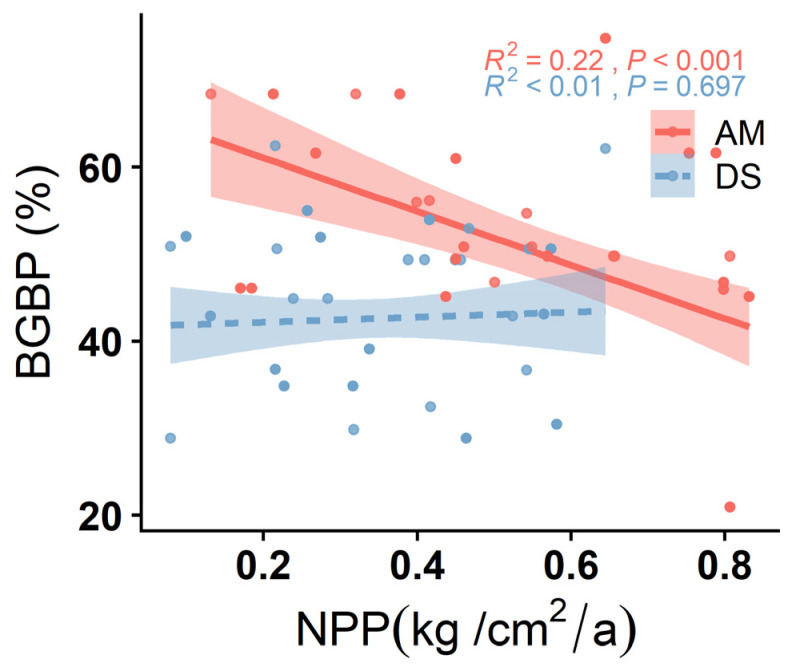
Linear relationship between net primary productivity (NPP) and BGBP (belowground biomass production) of alpine meadows and desert steppes. R^2^ represents the goodness of fit of the model, and *p*-values indicate significance.

**Figure 4 plants-13-00727-f004:**
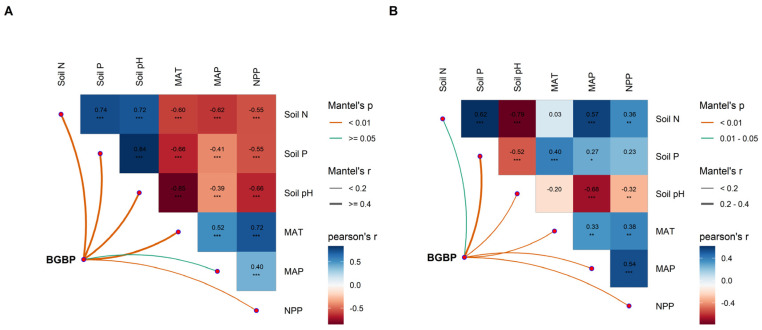
Multivariate correlation analysis of potential influencing factors in alpine meadows (**A**) and desert steppes (**B**), including climate factors (MAT and MAP), soil factors (Soil N, Soil P, and Soil pH), and net primary productivity (NPP). * *p* < 0.05; ** *p* < 0.01; *** *p* < 0.001.

**Figure 5 plants-13-00727-f005:**
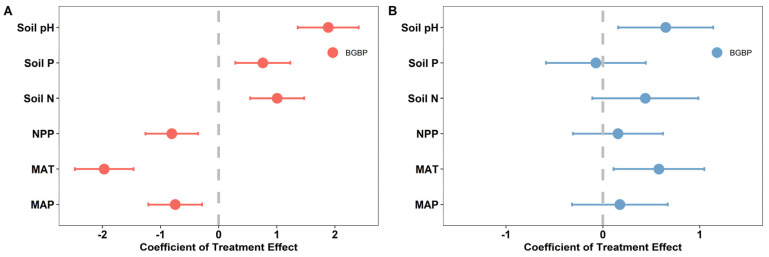
The high-level effects of each influencing factor on BGBP (belowground biomass production) in alpine meadows (**A**) and desert steppes (**B**). The numerical values represent the average effect size ± 95% confidence interval. The dotted line indicates an effect size of 0. The influencing factors are MAT, MAP, NPP, Soil N, Soil P, and Soil pH. We divided each influencing factor into two levels: high and low. The high level is the experimental group, which includes data values above the mean. The low level is the control group, which includes data values below the mean.

**Figure 6 plants-13-00727-f006:**
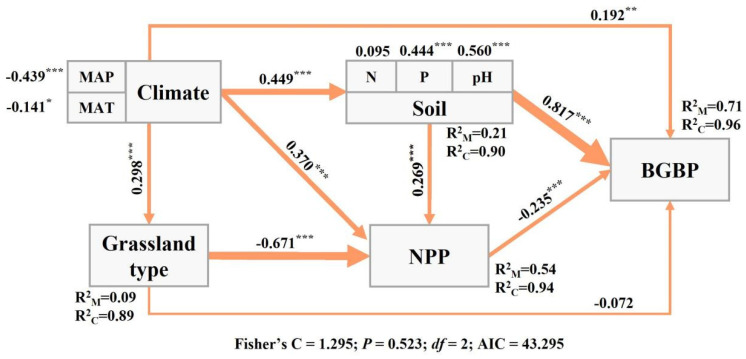
The relationships between climate factors, soil factors, NPP, and BGBP (belowground biomass production) in 182 different types of grasslands (alpine meadows and desert steppes) in China. The path diagram represents the standardized results of the final structural equation model (SEM) used to study the relationships between variables. The numbers next to the paths represent the standardized SEM coefficients, and asterisks indicate significance (*** *p* < 0.001; ** *p* < 0.01; * *p* < 0.05). R^2^ represents the goodness of fit of the generalized additive model (GAM). The best SEM model was selected based on the lowest AIC.

**Figure 7 plants-13-00727-f007:**
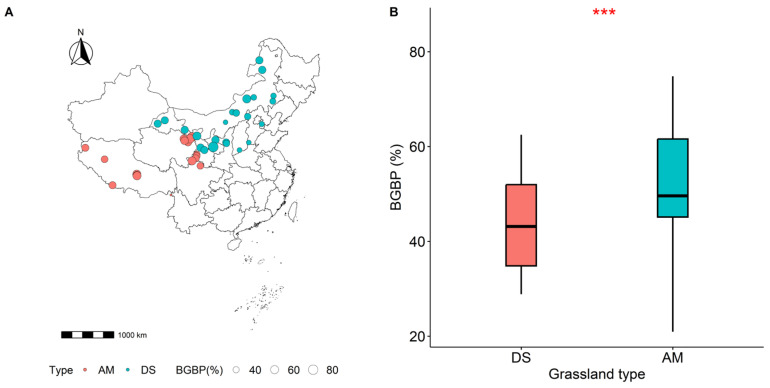
The spatial distribution and sample locations of alpine meadows (AMs) and desert steppes (DSs) in China (**A**). The comparison of belowground biomass production (BGBP) between alpine meadows and desert steppes is statistically significant at the 0.05 level (**B**). The *p*-value is less than 0.001 (*** *p* < 0.001).

## Data Availability

No new data were created or analyzed in this study. Data sharing is not applicable to this article.

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
