# Peer review of "Climate Factors Influence Above- and Belowground Biomass Allocations in Alpine Meadows and Desert Steppes through Alterations in Soil Nutrient Availability"

_plants, 2024, doi:10.3390/plants13050727_

Round 1
Reviewer 1 Report
Comments and Suggestions for Authors
I would like to thank you for your confidence in reviewing this manuscript.
I send you here my comments for the manuscript review.
Type of manuscript: Communication
Title: Climate factors affect above-belowground biomass allocation of grasslands by altering soil nutrient availability.
Comment: Could authors provide some photos, which present the experiments?
A conclusion section is missed.
Results:
Please check the Figures apparition, you should begin with Figure 1 for Figure 2, ………..
Materials and methods:
Figure 1 means Figure 7?
References:
Please check references (in text and list) in relation to the journal's recommendations. The reference list needs to be revised carefully.
Author Response
Reviewer 1(Remarks to the Author)
Comment: Could authors provide some photos, which present the experiments?
A conclusion section is missed.
Response: Thank you very much for your advice. I have provided some photos which present the experiments below.Fig.1 is a sample of desert grassland(DS). Fig.2 is a sample of alpine grassland(AM). We have also add the conclusion section.
Utilizing biomass data from 182 plots across 17 Alpine Meadows (AM) and 21 Desert Steppes (DS) in China, this study examined the influence of climate factors, soil nutrients, and Net Primary Productivity (NPP) on the BGBP across various grassland types. The findings revealed significant spatial distribution differences in BGBP among the grassland types. Climate-related factors, by modulating soil nutrient availability, emerged as the primary determinants of the BGBP spatial distribution patterns. Consequently, emphasizing the impact of climate change on the allocation of biomass above and below ground in grasslands is essential for forecasting terrestrial ecosystems' reactions to global climate change. Additionally, investigating the allocation strategies and influential factors on grassland plants' above- and below-ground biomass contributes to a deeper understanding of carbon allocation and storage dynamics within grassland ecosystems. This knowledge is crucial for accurately predicting carbon feedback on a regional scale in the future.
Fig.1
Fig.2
Results:
Please check the Figures apparition, you should begin with Figure 1 for Figure 2, ………..
Response:Thank you very much for your advice. We have readjusted the order of Figures 1 through 7.
Materials and methods:
Figure 1 means Figure 7?
Response: Thank you very much for your advice. We have updated the Figures order.
References:
Please check references (in text and list) in relation to the journal's recommendations. The reference list needs to be revised carefully.
Response:Thank you very much for your advice. We have carefully examined the format of the references and have revised them according to the format of the references in the Plants journal.

Reviewer 2 Report
Comments and Suggestions for Authors
The evaluated manuscript is interesting because the authors attempted to estimate the change in plant biomass, taking into account the root system in various growth and development conditions.
Line 265-266 The authors should indicate from what depth the roots were collected.
Generally Material and Methods chapter
line 276-281 The authors should provide information in the Material and Methods chapter on how they determined the content of nitrogen, phosphorus and pH in the soil? The authors should justify why the potassium content was not determined? In grasslands and midow, it is a much more important element than phosphorus. The links to the methods are in Chinese. Please include links that discuss the methods used in English.
No conclusions were found in the evaluated manuscript. This is one of the most important parts of any publication. Authors should provide conclusions as the last chapter of the manuscript.
Author Response
Reviewer 2(Remarks to the Author)
Line 265-266 The authors should indicate from what depth the roots were collected.
Response:Thank you very much for your advice. We have indicated the sampling depth in the resubmitted version of the article."The root system was collected by root excavation method in each sample plot, and the digging depth was 50cm".(Line.255-256)
line 276-281 The authors should provide information in the Material and Methods chapter on how they determined the content of nitrogen, phosphorus and pH in the soil? The authors should justify why the potassium content was not determined? In grasslands and midow, it is a much more important element than phosphorus. The links to the methods are in Chinese. Please include links that discuss the methods used in English.
Response:Thank you very much for your advice. The determination of the content of nitrogen, phosphorus and pH in the soil is provided in Line.269-271 of the newly submitted version of the article. The links are in Chinese, but to be honest I got the data from the linked websites. Nitrogen and phosphorus are the main nutrients necessary for plant growth, and they play a vital role in plant growth and development, so they have received extensive attention. In contrast, although potassium is also an essential nutrient for plants, its content in plants is relatively low, and its effect is not as direct and obvious as nitrogen and phosphorus, so our experiment did not measure potassium content. Many studies have shown that the limitations of nitrogen and phosphorus in soil physicochemical properties may be more common as to the limiting factors of biomass allocation in grassland ecosystems, so this study is more inclined to focus on the effects of nitrogen and phosphorus. Below is some literature on the effects of nitrogen and phosphorus on grassland biomass.
- Li, W.; Gan, X.; Jiang, Y.; Cao, F.; Lü, X.-T.; Ceulemans, T.; Zhao, C. Nitrogen Effects on Grassland Biomass Production and Biodiversity Are Stronger than Those of Phosphorus. Environ.Pollut.2022, 309, 119720.
- 2. Li, J.H.; Zhang, R.; Cheng, B.H.; Ye, L.F.; Li, W.J.; Shi, X.M. Effects of Nitrogen and Phosphorus Additions on Decomposition and Accumulation of Soil Organic Carbon in Alpine Meadows on the Tibetan Plateau. LDegrad.Dev. 2021, 32, 1467–1477.
- 3. Wang, Y.; Wang, C.; Ren, F.; Jing, X.; Ma, W.; He, J.-S.; Jiang, L. Asymmetric Response of Aboveground and Belowground Temporal Stability to Nitrogen and Phosphorus Addition in a Tibetan Alpine Grassland. Change Biol.2023, 29, 7072–7084.
No conclusions were found in the evaluated manuscript. This is one of the most important parts of any publication. Authors should provide conclusions as the last chapter of the manuscript.
Response:Thank you very much for your advice. We have added the conclusion section to the newly submitted version of the article.
Reviewer 3 Report
Comments and Suggestions for Authors
The manuscript concerns the issue of biomass production, on aboveground and underground parts of alpine meadows and desert steepees, which is interesting from an ecological point of view.
But from scientific point of view this issue of the manuscript are obvious and already documented by researches and taught in schools, hence I consider research on these issues to be secondary and not new.
Author Response
Reviewer 3(Remarks to the Author)
The manuscript concerns the issue of biomass production, on aboveground and underground parts of alpine meadows and desert steepees, which is interesting from an ecological point of view.
But from scientific point of view this issue of the manuscript are obvious and already documented by researches and taught in schools, hence I consider research on these issues to be secondary and not new.
Response:Thank you very much for your advice. Although some relevant theories have been mentioned in the textbook, I think my research is of practical significance. Compared with some small scale studies on biomass at the species level, this study quantified the spatial pattern of subsurface biomass allocation in grassland at the macro scale. At the same time, the results of this experiment can promote our understanding of the dynamics of carbon allocation and storage in grassland ecosystems, which is of great significance for carbon feedback prediction at regional scale in the future.
Reviewer 4 Report
Comments and Suggestions for Authors
The topic of the article is important bringing new information about factors affecting the belowground biomass of grasslands. Sites for data collection were chosen in natural grasslands with low affect of human impact, affected by extreme climate, but there is a large area of semi-natural grasslands under mild climate. As study is not bringing data from these stands, I would prefer to change the title of the manuscript not to make such an extensive generalisation of the results.
Abstract is informative, coherent with other parts of the manuscript.
Introduction covers most of the topics important to presented research. I only miss some details on already published data on aboveground/belowground biomass ratio.
Results are clearly presented and adequately statistically analysed. In figures only linear relationship is used – is there any other relationship that would fit better?
Discussion confronts results with other authors and it is sufficient.
Methods are described only briefly, I miss citations related to root excavation method and total harvest method, as they are only mentioned without any description. Which methods were used to quantify soil N, P content? Web page www.csdn.store can not be accessed.
Author Response
Reviewer 4(Remarks to the Author)
The topic of the article is important bringing new information about factors affecting the belowground biomass of grasslands. Sites for data collection were chosen in natural grasslands with low affect of human impact, affected by extreme climate, but there is a large area of semi-natural grasslands under mild climate. As study is not bringing data from these stands, I would prefer to change the title of the manuscript not to make such an extensive generalisation of the results.
Response:Thank you very much for your advice. We have modified the title according to your suggestion, and the revised title is as follows:
Climate Factors Influence Above- and Belowground Biomass Allocation in Alpine Meadows and Desert Steppes through Alterations in Soil Nutrient Availability
Abstract is informative, coherent with other parts of the manuscript.
Introduction covers most of the topics important to presented research. I only miss some details on already published data on aboveground/belowground biomass ratio.
Response:Thank you very much for your advice. In the introduction to the new submission, we have added some references to the published data on aboveground/belowground biomass ratio."The distribution of aboveground and below-ground biomass in grassland ecosystems is widely discussed, and in general, below-ground biomass is higher than above-ground biomass in grassland ecosystems[3,4]."(Line.38-41)
Results are clearly presented and adequately statistically analysed. In figures only linear relationship is used – is there any other relationship that would fit better?
Response:Thank you very much for your advice. In the figures in the result part, we compared the linear relationship and the nonlinear relationship in the previous preparation work to describe the relationship between each impact factor and BGBP, and the linear relationship is more suitable for the expression of the results of this study.
Discussion confronts results with other authors and it is sufficient.
Methods are described only briefly, I miss citations related to root excavation method and total harvest method, as they are only mentioned without any description. Which methods were used to quantify soil N, P content? Web page www.csdn.store can not be accessed.
Response:Thank you very much for your advice. We have described the root excavation method and total harvest method in more detail in the newly submitted methods section. "The root system was collected by root excavation method in each sample plot, and the digging depth was 50cm. When digging roots, care should be taken not to damage the roots, and the entire root system should be dug out as much as possible, including taproots, lateral roots and fine roots. The leaves and stems of the above-ground part of the plant are harvested by harvest method."(Line.256-259) In addition, there is no problem with the measurement method of soil N, P content and pH. The website cannot be opened because the website is under maintenance recently or for other reasons. soil N, P content and soil pH data from published in the journal (https://doi.org/10.5194/essd-13-5337-2021), and https://www.osgeo.cn/data/wc137.
- Zhang, Y.-W.; Guo, Y.; Tang, Z.; Feng, Y.; Zhu, X.; Xu, W.; Bai, Y.; Zhou, G.; Xie, Z.; Fang, J. Patterns of Nitrogen and Phosphorus Pools in Terrestrial Ecosystems in China. Earth System Science Data 2021, 13, 5337–5351, doi:10.5194/essd-13-5337-2021.
Round 2
Reviewer 2 Report
Comments and Suggestions for Authors
Line 254-256
Unfortunately, I did not find any information on how and by what methods the pH, N and P contents were determined. The authors provided literature in English, but unfortunately this information is not there. There is also no explanation why the authors did not determine the potassium content in the soil?
Author Response
Line 254-256
Unfortunately, I did not find any information on how and by what methods the pH, N and P contents were determined. The authors provided literature in English, but unfortunately this information is not there.
Response:Thank you very much for your advice. We supplemented the methods of soil pH, N and P contents determination. "Soil pH, N, and P within the uppermost 30 cm of soil were obtained from https://www.csdn.store and https://www.osgeo.cn/data/wc137. In the above data acquisition website, soil N concentrations of all samples were measured by a C–N analyser (PE-2400 II; Perkin-Elmer, Boston, USA), while soil P concentrations were measured using the molybdate–ascorbic acid method after H2SO4–H2O2 digestion [60]. The soil pH was determined in a 1:2.5 soil/water solution using a pH meter [61]."(Line.271-276)
There is also no explanation why the authors did not determine the potassium content in the soil?
Response:Thank you very much for your advice. Nitrogen and phosphorus are the main nutrients necessary for plant growth, and they play a vital role in plant growth and development, so they have received extensive attention. In contrast, although potassium is also an essential nutrient for plants, its content in plants is relatively low, and its effect is not as direct and obvious as nitrogen and phosphorus, so our experiment did not measure potassium content. Many studies have shown that the limitations of nitrogen and phosphorus in soil physicochemical properties may be more common as to the limiting factors of biomass allocation in grassland ecosystems, so this study is more inclined to focus on the effects of nitrogen and phosphorus [1-3].
- Li, W.; Gan, X.; Jiang, Y.; Cao, F.; Lü, X.-T.; Ceulemans, T.; Zhao, C. Nitrogen Effects on Grassland Biomass Production and Biodiversity Are Stronger than Those of Phosphorus. Environ.Pollut.2022, 309, 119720.
- 2. Li, J.H.; Zhang, R.; Cheng, B.H.; Ye, L.F.; Li, W.J.; Shi, X.M. Effects of Nitrogen and Phosphorus Additions on Decomposition and Accumulation of Soil Organic Carbon in Alpine Meadows on the Tibetan Plateau. LDegrad.Dev. 2021, 32, 1467–1477.
- 3. Wang, Y.; Wang, C.; Ren, F.; Jing, X.; Ma, W.; He, J.-S.; Jiang, L. Asymmetric Response of Aboveground and Belowground Temporal Stability to Nitrogen and Phosphorus Addition in a Tibetan Alpine Grassland. Change Biol.2023, 29, 7072–7084.
Reviewer 3 Report
Comments and Suggestions for Authors
Dear authors
I aknowleage your response to my general comment regarding the novelty and importance of the research undertaken. However, I will not change my opinion about the manuscript that I presented earlier.
I leave the decision about whether to publish the manuscript to the editor.